# Path Imputation Strategies for Signature Models

**Michael Moor** [1 2]   **Max Horn** [1 2]   **Christian Bock** [1 2]   **Karsten Borgwardt** [1 2]   **Bastian Rieck** [1 2]

## Abstract

The signature transform is a 'universal nonlinearity' on the space of continuous vector-valued paths, and has received attention for use in machine learning. However real-world temporal data is typically discretised, and must first be transformed into a continuous path before signature techniques can be applied. We characterise this as an imputation problem, and empirically assess the impact of various imputation techniques when applying signatures to irregular time series data. In our experiments, we find that the choice of imputation drastically affects shallow signature models, whereas deeper architectures are more robust. We also observe that uncertainty-aware *predictions* are overall beneficial, even compared to the uncertainty-aware *training* of Gaussian process (GP) adapters. Hence, we propose an extension of GP adapters by integrating uncertainty to the prediction step. This leads to competitive performance in general, and improves robustness in signature models in particular.

## 1. Introduction

Originally described by Chen [1954; 1957; 1958] and popularised in the theory of rough paths and controlled differential equations [Friz and Victoir, 2010; Lyons, 2014; 1998], the *signature transform*, also known as the *path signature* or simply *signature*, acts on a continuous vector-valued path of bounded variation, and returns a graded sequence of statistics, which determine a path up to a negligible equivalence class. Moreover, *every* continuous function of a path can be recovered by applying a linear transform to this collection of statistics [Bonnier et al., 2019, Proposition A.6].

[1]Department of Biosystems Science and Engineering, ETH Zurich, Switzerland [2]SIB Swiss Institute of Bioinformatics, Switzerland. Correspondence to: Michael Moor <michael.moor@bsse.ethz.ch>, Bastian Rieck <bastian.rieck@bsse.ethz.ch>.

*Presented at the first Workshop on the Art of Learning with Missing Values (Artemiss) hosted by the $37^{th}$ International Conference on Machine Learning (ICML). Copyright 2020 by the author(s).*

This 'universal nonlinearity' property makes the signature a promising nonparametric feature extractor with beneficial properties in both generative and supervised learning scenarios. Given their similarities, we may hope that tools that apply to continuous paths can *also* be extended to multivariate time series. But since multivariate time series are not continuous paths, one first needs to construct a continuous path to apply signature techniques. Previous work [Bonnier et al., 2019; Fermanian, 2019; Levin et al., 2013] characterise this construction as an embedding problem, and typically consider it a minor technical detail. We show that considering the path construction process is crucial for achieving competitive predictive performance: we reinterpret the task of constructing a continuous path, turning it from an embedding problem to an imputation problem, which we call *path imputation*. While previous work on the signature transform focussed on its excellent theoretical properties, such as sampling independence [Bonnier et al., 2019, Proposition A.7], our findings show that this does not necessarily correspond to empirical performance. However, since both observation rates and missingness itself are known to be informative for time series classification tasks [Rubin, 1976], we perform a thorough investigation of multiple imputation schemes in combination with various models that can potentially employ signatures. Moreover, since we observe that access to uncertainty information *during prediction* helps improve performance, we propose a novel extension to Gaussian process adapters [Li and Marlin, 2016], that is of independent interest. We make our code publicly available under `https://osf.io/ktc96/?view_only=62d41b4e60f64d49a3fb100a45d08116`.

## 2. Background

Let $f = (f_1, \ldots, f_d) \colon [a,b] \to \mathbb{R}^d$ be a continuous, piecewise differentiable path. Then the *signature transform up to depth $N$* is

$$\text{Sig}^N(f) = \left( (s_{i_1,\ldots,i_k})_{1 \leq i_1,\ldots,i_k \leq d} \right)_{1 \leq k \leq N}, \quad (1)$$

where each $s_{i_1,\ldots,i_k} \in \mathbb{R}$ is defined by

$$s_{i_1,\ldots,i_k} = \int \cdots \int_{a < t_1 < \cdots < t_k < b} \prod_{j=1}^{k} \frac{\mathrm{d}f_{i_j}}{\mathrm{d}t}(t_j) \, \mathrm{d}t_1 \cdots \mathrm{d}t_k. \quad (2)$$

This definition can be extended to paths of bounded variation by replacing these integrals with Stieltjes integrals with respect to each $f_{i_j}$. In brief, the signature transform may be interpreted as extracting information about *order* and *area* of a path. One may interpret its terms as 'the area/order of one channel with respect to some collection of other channels'. See Chevyrev and Kormilitzin [2016].

**Notation** Given a set $A$, we define the space of time series over $A$ by

$$
\mathcal{S}(A) = \{((t_1, x_1), \ldots, (t_n, x_n)) \mid t_i \in \mathbb{R}, x_i \in A, \\
n \in \mathbb{N}, \text{ s.t. } t_1 \leq \cdots \leq t_n\}. \quad (3)
$$

Furthermore, let $\mathcal{Y}$ be a set and let $\mathcal{X}_j = \mathbb{R}$ for $j \in \{1, \ldots, d\}$ and $d \in \mathbb{N}$. Then we assume that we observe a dataset of labelled time series $(\mathbf{x}_k, y_k)$ for $k \in \{1, \ldots, N\}$, where $\mathbf{x}_k \in \mathcal{S}(\mathcal{X}^*)$ and $y_k \in \mathcal{Y}$, with $\mathcal{X}^* = \prod_{j=1}^d (\mathcal{X}_j \cup \{*\})$ and $*$ representing no observation. We similarly define $\mathcal{X} = \prod_{j=1}^d \mathcal{X}_j$. Thus, $\mathcal{X}$ is the data space, while $\mathcal{X}^*$ is the data space allowing missing data, and $\mathcal{Y}$ is the set of labels.

**Gaussian process adapter** Some of the imputation schemes we consider are based on the uncertainty aware-framework of multi-task Gaussian process adapters [Futoma et al., 2017; Li and Marlin, 2016]. Let $\mathcal{W}, \mathcal{H}$ be some sets. Let $\ell: \mathcal{Y} \times \mathcal{Y} \to [0, \infty)$ be a loss function and $F: \mathcal{X}^{[a,b]} \times \mathcal{W} \to \mathcal{Y}$ be some (typically neural network) model, with $\mathcal{W}$ being a parameter space. Moreover, let $\mu: [a, b] \times \mathcal{S}(\mathcal{X}^*) \times \mathcal{H} \to \mathcal{X}$, $\Sigma: [a, b] \times [a, b] \times \mathcal{S}(\mathcal{X}^*) \times \mathcal{H} \to \mathcal{X}$ be mean and covariance functions, with $\mathcal{H}$ denoting hyperparameters. The dependence on $\mathcal{S}(\mathcal{X}^*)$ represents conditioning on observed values. Using Monte Carlo (MC) sampling, the goal is to solve

$$
\underset{\mathbf{w} \in \mathcal{W}, \boldsymbol{\eta} \in \mathcal{H}}{\arg\min} \sum_{k=1}^N \frac{1}{S} \sum_{s=1}^S \ell(F(\mathbf{z}_{s,k}, \mathbf{w}), y_k), \quad (4)
$$

with $\mathbf{z}_{s,k} \sim \mathcal{N}(\mu(\cdot, \mathbf{x}_k, \eta), \Sigma(\cdot, \cdot, \mathbf{x}_k, \eta))$.

## 3. Related work

A key motivation for this work is the use of the signature transform in machine learning: recent work [Chevyrev and Kormilitzin, 2016; Kormilitzin et al., 2016; Li et al., 2017; Morrill et al., 2019; Perez Arribas et al., 2018; Yang et al., 2016; 2017] typically employed the signature transform as a nonparametric feature extractor, on top of which a model is learnt. Integrations into typical neural networks are also being actively discussed [Bonnier et al., 2019; Liao et al., 2019; Reizenstein, 2019], as well as kernel-based approaches [Chevyrev and Oberhauser, 2018; Király and Oberhauser, 2019]. [Toth and Oberhauser, 2019] show how this kernel may be used to define a Gaussian process.

To our knowledge, no prior work has regarded path calculation as an *imputation problem*; typically, data is converted into a continuous path via linear/rectilinear interpolation [Chevyrev and Kormilitzin, 2016; Fermanian, 2019], or a hybrid of the two [Levin et al., 2013]. The general problem of imputing data is well-known and well-studied [Gelman and Hill, 2007, Chapter 25]. Imputation methods typically only fill in missing discrete data points, and do not attempt to impute the underlying continuous path. Gaussian process adapters [Li and Marlin, 2016], by contrast, are capable of imputing a *full* continuous path, from which we may sample arbitrarily. Hence, this framework also needs to be considered in this paper.

## 4. Path imputations for signature models

Signatures act on *continuous* paths. However, in real-world applications, temporal data typically appears as a discretised collection of measurements, potentially irregularly-spaced and asynchronously observed. To apply the signature to this data, it first has to be imputed into a continuous path. We believe this step to have a significant impact on the resulting signature, and thus also on models employing the signature. To assess this hypothesis, we explicitly treat this transformation as a *path imputation*, i.e. a mapping of the form $\phi: \mathcal{S}(\mathcal{X}^*) \to (\mathbb{R} \times \mathcal{X})^{[a,b]}$. We aim to learn a function $g: \mathcal{S}(\mathcal{X}^*) \to \mathcal{Y}$, which decomposes to $g = F \circ \phi$, where $F$ refers to a classifier, mapping from $(\mathbb{R} \times \mathcal{X})^{[a,b]} \times \mathcal{W}$ to $\mathcal{Y}$. Given a loss function $\ell$ and a set of $p$ path imputation strategies, $\Phi = (\phi_i)_{i=1}^p$, we minimise the objective:

$$
\underset{\phi_i \in \Phi, \mathbf{w} \in \mathcal{W}}{\arg\min} \quad \mathbb{E}_{(\mathbf{x},y) \sim P(\mathcal{S}(\mathcal{X}^*), \mathcal{Y})} \left[ \ell(g(\mathbf{x}; \phi_i, \mathbf{w}), y) \right] \quad (5)
$$

Even though Equation 5 could be formulated more *implicitly* (i.e. without any explicit imputation step), this formulation enables us to make explicit how the signature transform 'interprets' the raw data for downstream classification tasks.

**Path imputation strategies** We consider the following set of strategies for path imputation, i.e. (1) linear interpolation, (2) forward filling, (3) indicator imputation, (4) zero imputation, (5) causal imputation[1], and (6) Gaussian process adapters. Strategies 1–5 can be seen as a fixed preprocessing step, whereas GP adapters (strategy 6) are optimised end-to-end with the downstream task. For more details regarding these strategies, please refer to Section A.3 in the appendix.

For conventional GP adapters, one major drawback with the formulations of Li and Marlin [2016] and Futoma et al. [2017], as described in equation (4), is that approximating the expectation outside of the loss function with MC sampling is expensive. During prediction, Li and Marlin [2016]

---

[1]This strategy is similar to the time-joined transformation [Levin et al., 2013]. For more details, please refer to Section A.6 in the appendix.

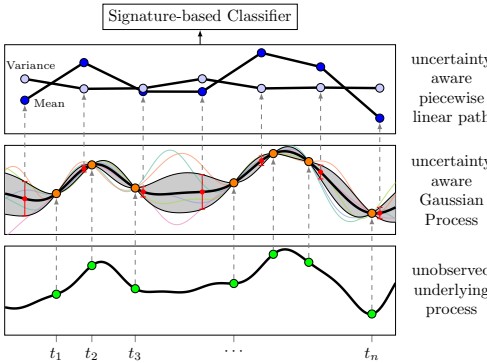

Figure 1: Overview of our proposed extension of GP adapters, GP-PoM, leveraging both posterior moments (mean and variance)

proposed to overcome this issue by sacrificing the uncertainty in the loss function and to simply pass the posterior mean, as in supplementary Equation (12)[2]. To address both points, we propose to instead also pass the posterior covariance of the Gaussian process to the classifier $F$. This saves the cost of MC sampling whilst explicitly providing $F$ with uncertainty information during the prediction[3]. However, the full covariance matrix may become very large, and it is not obvious that all interactions are relevant to the subsequent classifier. This is why we simplify matters by taking the posterior *variance* at every point, and concatenate it with the posterior mean at every point, to produce a path whose evolution also captures the uncertainty at every point:

$$\tau \colon [a,b] \times \mathcal{S}(\mathcal{X}^*) \times \mathcal{H} \to \mathcal{X} \times \mathcal{X} \qquad (6)$$

$$\tau \colon t, \mathbf{x}, \eta \mapsto (\mu(t, \mathbf{x}, \eta), \Sigma(t, t, \mathbf{x}, \eta)). \qquad (7)$$

This corresponds to solving

$$\underset{\mathbf{w} \in \mathcal{W}, \boldsymbol{\eta} \in \mathcal{H}}{\arg\min} \sum_{k=1}^{N} \ell(F(\tau(\,\cdot\,, \mathbf{x}_k, \eta), \mathbf{w}), y_k), \qquad (8)$$

where instead now $F \colon (\mathcal{X} \times \mathcal{X})^{[a,b]} \times \mathcal{W} \to \mathcal{Y}$. Notice that when $F$ is a signature model, it is now straightforward to compute the signature of the Gaussian process, simply by querying many points to construct a piecewise linear approximation to the process. Figure 1 depicts our proposed strategy GP-PoM.

## 5. Experiments

We first introduce our experimental setup (datasets and model architectures) before presenting and discussing quantitative results.

---

[2]Equations (4) and (12) are of course not in general equal, so following Futoma et al. [2017], our standard GP adapter uses MC sampling both in training and testing.

[3]Even if MC sampling is used during prediction, $F$ has no per-sample access to uncertainty about the imputation.

**Datasets & preprocessing**  We analyse four real-world time series datasets, i.e. (i) `Physionet2012` [Goldberger et al., 2000], (ii) `PenDigits` [Dua and Graff, 2017], (iii) `LSST` [Allam Jr et al., 2018], and (iv) `CharacterTrajectories` [Dua and Graff, 2017]. Moreover, to efficiently compute the signature, we sample the imputed path in a *fixed* time resolution, resulting in a piecewise linear path. For time series that are not irregularly spaced, we apply two types of random subsampling as an additional preprocessing step for all but the `Physionet2012` dataset, namely (1) 'Random': Missing at random; on the instance level, we discard 50% of all observations. (2) 'Label-based': Missing not at random; for each class, we uniformly sample missingness frequencies between 40% and 60%. Since `PenDigits` consists of particularly short time series (8 steps, 2 dimensions), we use more moderate frequencies of 30% and 20–40%, respectively, for discarding observations.

**Models**  We study the following models: (1) Sig, a simple signature model that involves a linear augmentation, the signature transform (signature block) and a final module of dense layers, (2) RNN, an RNN model using GRU cells [Cho et al., 2014], (3) RNNSig, which extends the signature transform to a window-based stream of signatures, and where the final neural module is a GRU sliding over the stream of signatures, and (4) DeepSig, a deep signature model sequentially employing two signature blocks featuring augmentation and signature transforms, following Bonnier et al. [2019]. Please refer to Supplementary Section A.4 for more details about the architectures and implementations. We use the 'Signatory' package to calculate the signature transform [Kidger and Lyons, 2020], and implemented all GP adapters in the 'GPyTorch' framework [Gardner et al., 2018].

**Training and evaluation**  We use the predefined training and testing splits for each dataset, separating 20% of the training split as a validation set for hyperparameter tuning. For each setting, we run a randomised hyperparameter search of 20 calls and train each of these fits until convergence (at most 100 epochs; we stop early if the performance on the validation split does not improve for 20 epochs). As for performance metrics, for binary classification tasks, we use average precision, for multi-class classification we focus on balanced accuracy (BAC). Having selected the best hyperparameter configuration for each setting, we repeat 5 fits, per fit select the best model state in terms of the best validation performance, and finally report mean and standard deviation (error bars) of the performance metrics on the testing split.

**Results**  Figure 2 depicts the results for the `Character Trajectories` dataset (please refer to supplemental Fig-

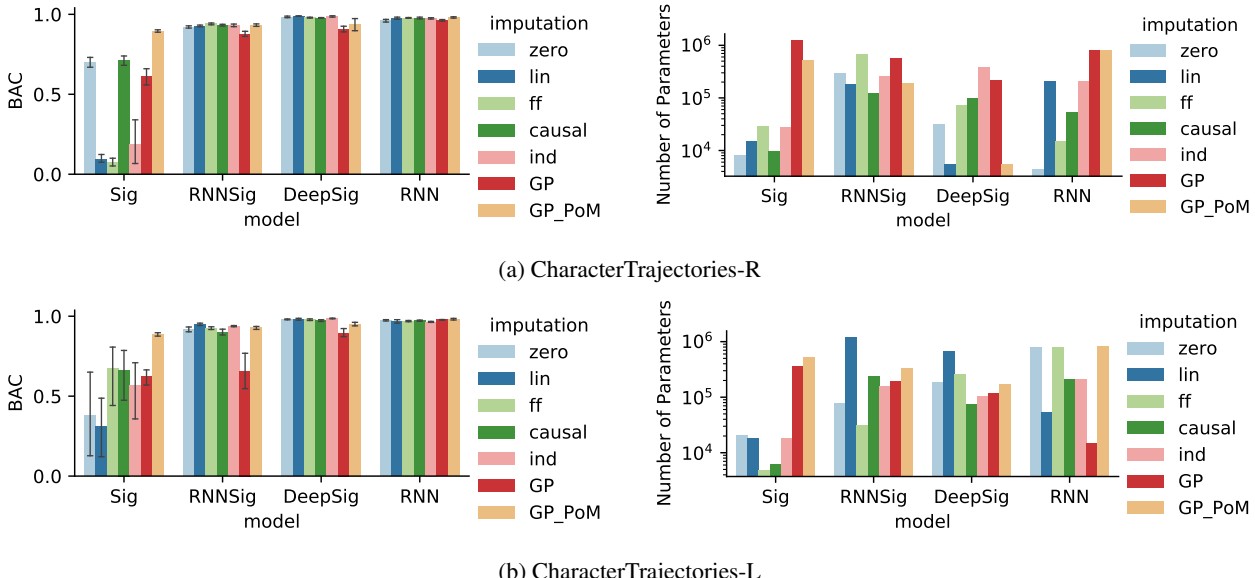

(a) CharacterTrajectories-R

(b) CharacterTrajectories-L

Figure 2: Experimental Results for Character Trajectories Dataset. The rows indicate the subsampling type: Random (R) versus Label-based (L). The left column displays performance in terms of balanced accuracy (BAC), whereas the right columns show the number of trainable parameter of the best model.

ure 3 for the other datasets). We observe that both DEEPSIG as well as the signature-free RNN perform well over many scenarios. In particular, they are robust to various imputation schemes. However, we also see that certain signature models, in particular SIG, are heavily impacted by the choice of imputation strategy. In the case of Character Trajectories, SIG was only able to achieve acceptable performance through our novel GP-POM strategy. In PenDigits, we encountered issues of numerical stability for the original GP adapter[4]; not so for GP-POM.

aware framework'), since for each MC sample, the GP adapter model has no access to missingness or uncertainty about the underlying imputation. GP-POM, our proposed end-to-end imputation strategy, shows competitive classification performance, while considerably improving upon the existing GP adapter. As for limitations, GP-POM sacrifices the GP adapter's ability to be explicitly uncertain *about* its own prediction (due to the variance of the MC sampled predictions), while the subsequent classifier has to be able to handle the doubled feature dimensionality.

## 6. Discussion

Our findings suggest that the choice of path imputation strategy can *drastically* affect the performance of signature-based models. We observe this most prominently in 'shallow' signature models. Among signature models, we found that deep signature models (DEEPSIG) are most robust in tackling irregular time series over different imputations—comparable to non-signature RNNs, yet on average being more parameter-efficient.

Overall, we find that uncertainty-aware approaches (indicator imputation and GP-POM) are beneficial when imputing irregularly-spaced time series for classification. Crucially, uncertainty information has to be accessible during the *prediction step*. We find that this is indeed not the case for the standard GP adapter (despite the naming of 'uncertainty-

## 7. Conclusion

The signature transform has recently gained attention for being a promising feature extractor that can be easily integrated to neural networks. As we empirically demonstrated in this paper, the application of signature transforms to real-world temporal data is fraught with pitfalls—specifically, we found the choice of an imputation scheme to be crucial for obtaining high predictive performance. Moreover, by integrating uncertainty to the prediction step, our proposed GP-POM has demonstrated overall competitive performance and in particular improved robustness in signature models when dealing with irregularly-spaced and asynchronous time series.

---

[4]They were addressed by jittering the diagonal in the Cholesky decomposition.

## Acknowledgements

This project was supported by the grant #2017-110 of the Strategic Focal Area 'Personalized Health and Related Technologies (PHRT)' of the ETH Domain for the SPHN/PHRT Driver Project 'Personalized Swiss Sepsis Study' (M.M., M.H., and C.B.; grant awarded to K.B.) and the Alfried Krupp Prize for Young professors of the Alfried Krupp von Bohlen und Halbach-Stiftung (K.B.). The authors are grateful to Patrick Kidger for valuable discussions, guidance, and code contributions.

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

# A. Appendix

## A.1. Further Experimental Results

See Figure 3.

## A.2. Gaussian process adapter

Some of the imputation schemes we consider are based on the uncertainty aware-framework of multi-task Gaussian process adapters [Futoma et al., 2017; Li and Marlin, 2016]. Hence, this section provides a more thorough description of the method. Let $\mathcal{W}, \mathcal{H}$ be some sets. Let $\ell \colon \mathcal{Y} \times \mathcal{Y} \to [0, \infty)$ be a loss function. Let $F \colon \mathcal{X}^{[a,b]} \times \mathcal{W} \to \mathcal{Y}$, be some (typically neural network) model, with $\mathcal{W}$ interpreted as a space of parameters. Moreover, let $\mu \colon [a,b] \times \mathcal{S}(\mathcal{X}^*) \times \mathcal{H} \to \mathcal{X}$ $\Sigma \colon [a,b] \times [a,b] \times \mathcal{S}(\mathcal{X}^*) \times \mathcal{H} \to \mathcal{X}$ be mean and covariance functions, with $\mathcal{H}$ representing hyperparameters. The dependence on $\mathcal{S}(\mathcal{X}^*)$ represents conditioning on observed values. Then the goal is to solve

$$\overbrace{\mathbb{E}_{\mathbf{z}_k \sim \mathcal{N}(\mu(\cdot, \mathbf{x}_k, \eta), \Sigma(\cdot, \cdot, \mathbf{x}_k, \eta))} \left[ \ell(F(\mathbf{z}_k, \mathbf{w}), y_k) \right]}^{E_k}. \quad (9)$$

As this expectation is typically not tractable, it is estimated by Monte Carlo (MC) sampling with $S$ samples, i.e.

$$E_k \approx \frac{1}{S} \sum_{s=1}^{S} \ell(F(\mathbf{z}_{s,k}, \mathbf{w}), y_k), \quad (10)$$

where

$$\mathbf{z}_{s,k} \sim \mathcal{N}\left(\mu(\cdot, \mathbf{x}_k, \eta), \Sigma(\cdot, \cdot, \mathbf{x}_k, \eta)\right). \quad (11)$$

Alternatively, one may forgo allowing the uncertainty to propagate through $F$ by instead passing the posterior mean directly to $F$; this corresponds to solving

$$\underset{\mathbf{w} \in \mathcal{W}, \boldsymbol{\eta} \in \mathcal{H}}{\arg\min} \sum_{k=1}^{N} \ell(F(\mu(\cdot, \mathbf{x}_k, \eta), \mathbf{w}), y_k). \quad (12)$$

## A.3. Imputation strategies

We consider the following set of strategies for path imputation, i.e.

1. linear interpolation: At a given imputation point, the previous and next observed data point are linearly interpolated.

2. forward filling: At a given imputation point, the last observed value is carried forward.

3. indicator imputation: At a given imputation point, for each feature dimension, if no observation is available an binary missingness indicator variable is set to 1, 0 otherwise. The missing value is filled with 0.

4. zero imputation: At a given imputation point, missing values are filled with 0.

5. causal imputation: This approach is related to forward filling and motivated by signature theory. As opposed to forward filling, the time and the actual value are updated sequentially. For more details, we introduce causal imputation in Section A.6.

6. Gaussian process adapter: We introduce GP adapters in Section A.2, where $\mathbf{z}$ refers to the imputed time series (modelled as Gaussian distribution).

## A.4. Model implementations, architectures and hyperparameters

All models are implemented in Pytorch [Paszke et al., 2019], whereas the GP adapter and GP-POM are implemented using the GPyTorch framework [Gardner et al., 2018]. Next, we specify the details of the model architectures.

**SIG** We use a simple signature model that involves one signature block comprising of a linear augmentation followed by the signature transform. Subsequently, a final module of dense layers $(30, 30)$ is used. This is architecture refers to the Neural-signature-augment model [Bonnier et al., 2019].

**RNNSIG** This model extends the signature transform to a window-based stream of signatures, where the final neural module is a GRU sliding over the stream of signatures. We allowed window sizes between 3 and 10 steps. For the GRU cell, we allowed any of the following number of hidden units: $[16, 32, 64, 128]$.

**RNN** Here, we use a standard RNN model using GRU cells. The size of hidden units was chosen as one of the following: $[16, 32, 64, 128, 256, 512]$.

**DEEPSIG** For the deep signature model we employ two signature blocks (each comprising a linear augmentation and the signature calculation) following Bonnier et al. [2019].

### Hyperparameters

For all signature-based models, we allowed a signature truncation depth of 2–4, as we observed that larger values quickly led to a parameter explosion. All models were optimized using Adam [Kingma and Ba, 2014]. Both the learning rates and weight decay were drawn log-uniformly between $10^{-4}$ and $10^{-2}$. We allowed for the following batch-sizes: $(32, 64, 128, 256)$. For GP-based models, to save memory, we used virtual batching based on a batch-size of 32. Furthermore, all GP-models employ an RBF-kernel and are implemented as Hadamard multi-task Gaussian process.

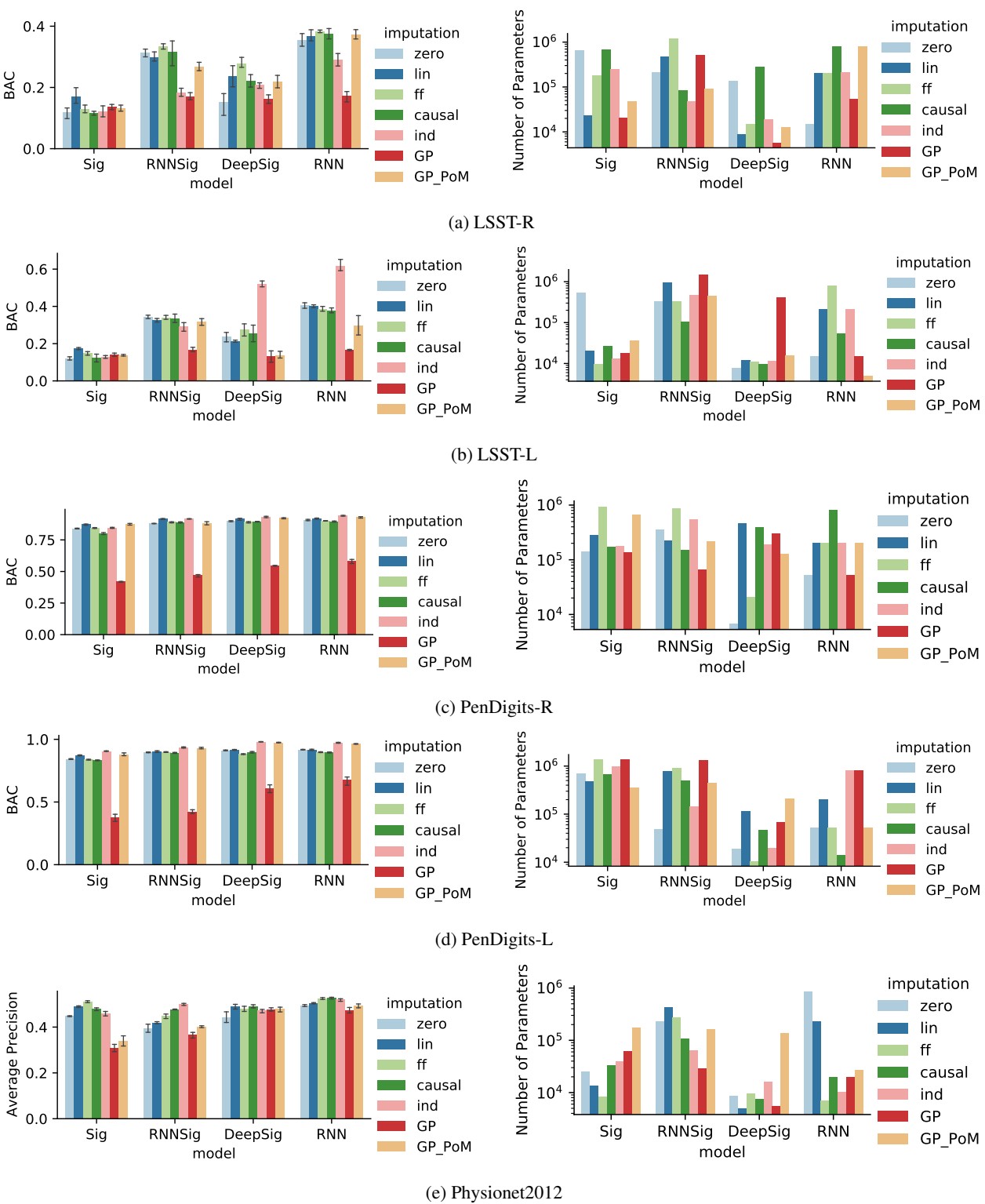

(a) LSST-R

(b) LSST-L

(c) PenDigits-R

(d) PenDigits-L

(e) Physionet2012

Figure 3: Further Experimental results. The rows indicate datasets and different subsampling schemes (R for Random, L for Label-based). The left column displays the performance metric which was optimzied for: balanced accuracy (BAC), or average precision. The right column indicates the number of trainable parameters for the selected models.

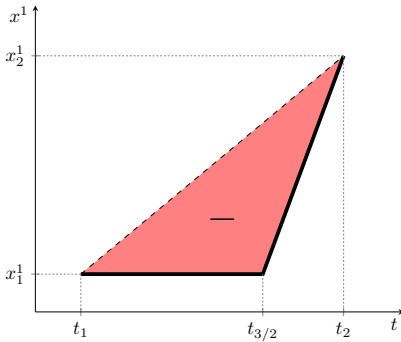

Figure 4: Lévy area of the forward-fill imputed path. By changing $t_{3/2}$ (a *single* unrelated observation!), we can make this disparity greater or smaller.

## A.5. Fragile dependence on sampling in unrelated channels: example

Suppose that we have observed the (very short) time series

$$\mathbf{x} = ((t_1, x_1^1, x_1^2), (t_2, x_2^1, *)) \in \mathcal{S}(\mathbb{R}^2). \qquad (13)$$

Perhaps we now apply, say, forward fill data-imputation, to produce

$$((t_1, x_1^1, x_1^2), (t_2, x_2^1, x_1^2)).$$

Finally we linearly path-impute to create the linear path

$$f \colon [t_1, t_2] \to \mathbb{R} \times \mathbb{R}^2$$
$$f \colon t \mapsto \left( t, x_1^1 \frac{t_2 - t}{t_2 - t_1} + x_2^1 \frac{t - t_1}{t_2 - t_1}, x_1^2 \right),$$

to which we may then apply the signature transform. In particular we will have computed the Lévy area with respect to $t$ and $x^1$. As this is just a straight line, the Lévy area is zero.

Now suppose we include an additional observation at some time $t_{3/2} \in (t_1, t_2)$, so that our data is instead

$$\mathbf{x} = ((t_1, x_1^1, x_1^2), (t_{3/2}, *, x_{3/2}^2), (t_2, x_2^1, *)). \qquad (14)$$

Then the same procedure as before will produce the data

$$\mathbf{x} = ((t_1, x_1^1, x_1^2), (t_{3/2}, x_1^1, x_{3/2}^2), (t_2, x_2^1, x_{3/2}^2)),$$

with corresponding function $f$. The $(t, x^1)$ components of $f$ and its $(t, x^1)$-Lévy area are shown in Figure 4. As a result of an unrelated observation in the $x^2$ channel, the $(t, x^1)$-Lévy area has been changed. The closer $t_{3/2}$ is to $t_2$, the greater the disparity. This simple example underscores the danger of 'just forward-fill data-imputing'. Doing so has introduced an undesired dependency on the simple *presence* of an observation in other channels, with the change in our imputed path being determined by the *time* at which this other observation occurred.

Indeed, *any* imputation scheme that predicts something other than the unique value lying on the dashed line in Figure 4, will fail. This means that this example holds for essentially every data-imputation scheme—the only scheme that survives this flaw is the linear data-imputation scheme. This is the unique imputation scheme that coincides with the linear path-imputation that *must* be our concluding step. However, when there is missing data at the start or the end of a partially observed times series, then there is no 'next observation' which linear imputation may use. So in general, we cannot uniformly apply the linear data-imputation scheme, and must choose another scheme.

## A.6. Causal signature imputation

In Section A.5 we have spoken about the limitations of traditional data-imputation schemes, and at first glance one may be forgiven for thinking that these are issues are unavoidable. However, it turns out that we need not be limited just to these traditional imputation schemes. The trick is to consider time not as a *parameterisation*, but as a *channel*[5]. This leads to a 'meta imputation strategy', which we refer to as *causal signature imputation*. It will turn any traditional causal data-imputation strategy (for example, feed-forward) into a causal path-imputation strategy for signatures; at the same time it will overcome the issue of a fragile dependence.

Suppose we have $\mathbf{x} \in \mathcal{S}(\mathcal{X}^*)$, and some favourite choice of causal data-imputation strategy $c \colon \mathcal{S}(\mathcal{X}^*) \to \mathcal{S}(\mathcal{X})$. Next, given

$$\mathbf{x} = ((t_1, x_1), \ldots, (t_n, x_n)) \in \mathcal{S}(\mathcal{X}), \qquad (15)$$

we define the operation $\Omega \colon \mathcal{S}(\mathcal{X}) \to \mathcal{S}(\mathcal{X})$ by

$$\Omega(\mathbf{x}) = ((t_1, x_1), (t_2, x_1), (t_2, x_2), (t_3, x_2),$$
$$\ldots,$$
$$(t_i, x_i), (t_{i+1}, x_i), (t_{i+1}, x_{i+1}), (t_{i+2}, x_{i+1}),$$
$$\ldots,$$
$$(t_{n-1}, x_{n-1}), (t_n, x_{n-1}), (t_n, x_n)). \qquad (16)$$

That is, *first* time is updated, and *then* the corresponding observation in data space is updated. This means that the change in data space occurs instantaneously.

For each $n \in \mathbb{N}$ (and given $a < b$), fix any $s_i^{(n)}$ for $i \in \{1, \ldots, n\}$. (We will see that the exact choice is unimportant in a moment.) Given

$$\mathbf{x} = ((t_1, x_1), \ldots, (t_n, x_n)) \in \mathcal{S}(\mathcal{X}),$$

---

[5] To be clear, using time as a channel is already a well-known trick in the signature literature that we do not take credit for inventing! See for example Bonnier et al. [2019, Definition A.3]. It is however pleasing that something commonly used in the theory of signatures is also what allows us to overcome what we identify as some of their limitations.

let $\psi\colon \mathcal{S}(\mathcal{X}) \to (\mathbb{R} \times \mathcal{X})^{[a,b]}$ be the unique continuous piecewise linear path such that $\psi(s_i^{(n)}) = (t_i, x_i)$. Note that this is just a slight generalisation of the linear path-imputation that has already been performed so far; we are simply no longer asking for additional assumptions of the form $s_i^{(n)} = t_i$.[6]

Finally, we put this all together, and define the causal signature imputation strategy $\phi_c$ associated with $c$ to be

$$\phi_c = \psi \circ \Omega \circ c,$$

which will be a map $\mathcal{S}(\mathcal{X}^*) \to (\mathbb{R} \times \mathcal{X})^{[a,b]}$. Thus $\phi_c$ defines a family of path-imputation schemes, parameterised by a choice of data-imputation scheme.

Before we analyse *why* this works in practice, we repeat a crucial property of the signature transform [Bonnier et al., 2019, Appendix A].

**Theorem 1** (Invariance to reparameterisation). *Let $f\colon [a,b] \to \mathbb{R}^d$ be a continuous piecewise differentiable path. Let $\psi\colon [a,b] \to [c,d]$ be continuously differentiable, increasing, and surjective. Then $\mathrm{Sig}^N(f) = \mathrm{Sig}^N(f \circ \psi)$.*

Coming back to our analysis, we first note that the previous theorem implies that the signature transform of $\phi_c(\mathbf{x})$ is invariant to the choice of $s_i^{(n)}$. Second, note that holding time between observations fixed is a valid choice, by the definition for $\mathcal{S}$ in equation (3). There should hopefully be no moral objection to our definition of $\mathcal{S}$, as holding time fixed essentially just corresponds to a jump discontinuity; not such a strange thing to have occur. Here, by replacing time as the parameterisation, we are then able to recover the continuity of the path. Third, we claim that $\phi_c$ is immune to the two major flaws of imputation methods, namely (i) their fragile dependence on sampling in unrelated channels, and (ii) their non-causality. Let us consider the first flaw of dependence on sampling in unrelated channels. For simplicity, take $c$ to be the forward-fill data-imputation strategy. Consider again the $\mathbf{x}$ defined in expression (13). This means that

$$\phi_c(\mathbf{x}) = \psi(\,((t_1, x_1^1, x_1^2), (t_2, x_1^1, x_1^2), (t_2, x_2^1, x_1^2))\,). \tag{17}$$

Contrast adding in the extra observation at $t_{3/2}$ as in equation (14). Then

$$\begin{aligned}
\phi_c(\mathbf{x})(s) \\
= \psi(\,&((t_1, x_1^1, x_1^2), (t_{3/2}, x_1^1, x_1^2), (t_{3/2}, x_1^1, x_{3/2}^2), \\
&(t_2, x_1^1, x_{3/2}^2), (t_2, x_2^1, x_{3/2}^2))\,).
\end{aligned} \tag{18}$$

Evaluating each $\psi$ will then in each case give a path with three channels, corresponding to $t, x^1, x^2$. Then it is clear

that the $(t, x^1)$ component of the path in equation (17) is just a reparameterisation of the path in equation (18), a difference which is irrelevant by Theorem 1. (And the $x^2$ component of the second path has been updated to use the new information $x_{3/2}^2$.) Thus the causal path imputation scheme is robust to such issues. For general time series and $c$ taken to be any other causal data-imputation strategy, then much the same analysis can be easily be performed.

Now consider the second potential flaw, of non-causality. The issue previously arose because of the non-causality of the linear path-imputation. We see from equation (16), however, such changes only occur in data space while the time channel is frozen; conversely the time channel only updates with the value in the data space frozen. Provided that $c$ is also causal, then causality will, overall, have been preserved. For example, it is possible to use this scheme in an online setting. There are interesting comparisons to be made between causal signature imputation and certain operations in the signature literature. First is the *lead-lag* transform [Chevyrev and Kormilitzin, 2016]. With the lead-lag transform, the entire path is *duplicated*, and then each side is alternately updated. Conversely, in causal signature imputation, the path is instead *split* between $t$ and $(x^1, \ldots, x^n)$, and then each side is alternately updated. Second is the comparison to the linear and rectilinear embedding strategies, see for example [Fermanian, 2019]. It is possible to interpret $\psi \circ \Phi$ as a hybrid between the linear and rectilinear embeddings: it is rectilinear with respect to an ordering of $t$ and $(x^1, \ldots, x^n)$, and linear on $(x^1, \ldots, x^n)$. Furthermore, the time-joined transformation [Levin et al., 2013] is pursuing a very similar goal to the here described causal signature imputation. This is also why we do not consider this imputation strategy as a novel contribution of this work.

---

[6]As in the $\varphi_\theta$ of [Toth and Oberhauser, 2019], for example.