# OpenReview forum: "Path Imputation Strategies for Signature Models"
_ICML.cc/2020/Workshop/Artemiss — ICML Artemiss 2020_

### Official Review · AnonReviewer1 · 2020-06-20
**Interesting contirbution that poses interpolation as path imputation showing it to be useful when performing signature transforms**

**Rating:** 8
**Confidence:** 3

**Review:**

Summary:
Discrete time series interpolation to obtain continuous paths is treated as a path imputation problem. The work proposes Gaussian process adapter based imputation strategy that shows improved performance on shallow  signature transform models.

Strengths:
+ This is an interesting contribution that could be applied to several time series problems where feature extraction using signature transforms could be used.
+ The paper is clearly written
+ Experiments are well done and highlight the contributions

Weaknesses:
- However, presentation and discussion of the results are unconvincing. Lacks clarity.
- A plot comparing the performance of different models with the proposed GP-PoM imputation strategy could have been useful. This information is present in Figure 2 but a focused discussion can be useful.
- Is the conclusion that GP-PoM useful only when using shallow models (like SIG)? I think this is mentioned but not tied to the presented results.
- There is no discussion surrounding the figure with number of parameters making it hard to appreciate. Why does the DeepSig model with GP-PoM model fewer parameters than when using other imputations?

---

### Decision · Program_Chairs · 2020-07-02

**Decision:**

Accept

**Comment:**

We're happy to accept this paper at Artemiss. We'll contact you soon to inform you about more details concerning the format of your presentation at the workshop, and the camera-ready version deadline. Please take into account the referee's comments to write the camera-ready version.